# INDUCTIVE REPRESENTATION LEARNING ON TEMPORAL GRAPHS

**Da Xu**[*]**, Chuanwei Ruan**[*]**, Evren Korpeoglu , Sushant Kumar , Kannan Achan**
Walmart Labs
Sunnyvale, CA 94086, USA
`{Da.Xu,Chuanwei.Ruan,EKorpeoglu,SKumar4,KAchan}@walmartlabs.com`

## ABSTRACT

Inductive representation learning on temporal graphs is an important step toward salable machine learning on real-world dynamic networks. The evolving nature of temporal dynamic graphs requires handling new nodes as well as capturing temporal patterns. The node embeddings, which are now functions of time, should represent both the static node features and the evolving topological structures. Moreover, node and topological features can be temporal as well, whose patterns the node embeddings should also capture. We propose the temporal graph attention (TGAT) layer to efficiently aggregate temporal-topological neighborhood features as well as to learn the time-feature interactions. For TGAT, we use the self-attention mechanism as building block and develop a novel functional time encoding technique based on the classical Bochner's theorem from harmonic analysis. By stacking TGAT layers, the network recognizes the node embeddings as functions of time and is able to *inductively* infer embeddings for both new and observed nodes as the graph evolves. The proposed approach handles both node classification and link prediction task, and can be naturally extended to include the temporal edge features. We evaluate our method with *transductive* and *inductive* tasks under temporal settings with two benchmark and one industrial dataset. Our TGAT model compares favorably to state-of-the-art baselines as well as the previous temporal graph embedding approaches.

## 1 INTRODUCTION

The technique of learning lower-dimensional vector embeddings on graphs have been widely applied to graph analysis tasks (Perozzi et al., 2014; Tang et al., 2015; Wang et al., 2016) and deployed in industrial systems (Ying et al., 2018; Wang et al., 2018a). Most of the graph representation learning approaches only accept static or non-temporal graphs as input, despite the fact that many graph-structured data are time-dependent. In social network, citation network, question answering forum and user-item interaction system, graphs are created as temporal interactions between nodes. Using the final state as a static portrait of the graph is reasonable in some cases, such as the protein-protein interaction network, as long as node interactions are timeless in nature. Otherwise, ignoring the temporal information can severely diminish the modelling efforts and even causing questionable inference. For instance, models may mistakenly utilize future information for predicting past interactions during training and testing if the temporal constraints are disregarded. More importantly, the dynamic and evolving nature of many graph-related problems demand an explicitly modelling of the timeliness whenever nodes and edges are added, deleted or changed over time.

Learning representations on temporal graphs is extremely challenging, and it is not until recently that several solutions are proposed (Nguyen et al., 2018; Li et al., 2018; Goyal et al., 2018; Trivedi et al., 2018). We conclude the challenges in three folds. **Firstly**, to model the temporal dynamics, node embeddings should not be only the projections of topological structures and node features but also functions of the continuous time. Therefore, in addition to the usual vector space, temporal representation learning should be operated in some functional space as well. **Secondly**, graph topological structures are no longer static since the nodes and edges are evolving over time, which poses

---

[*]Both authors contributed equally to this research.

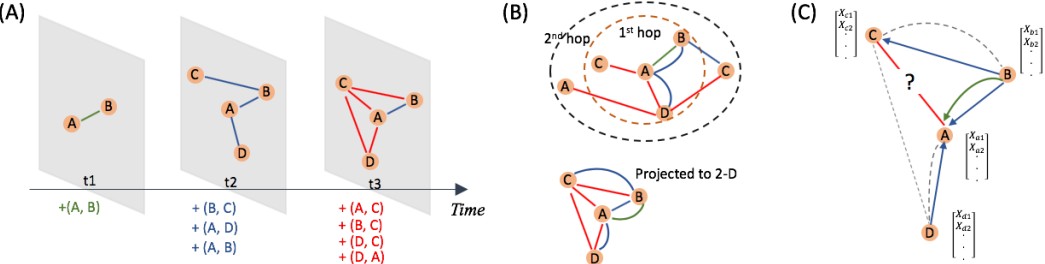

Figure 1: Visual illustration for several complications from the temporal graphs. (A). The generation process of a temporal graph and its snapshots. It is obvious that the static graphs in the snapshots only reflect partial temporal information. (B). The final state of the temporal graph when projected to the time-independent 2-D plane. Other than the missing temporal information, the multi-edge situation arises as well. (C). When predicting the link between node A and C at time $t_3$, the message-passing paths should be subject to temporal contraints. The solid lines give the appropriate directions, and the dashed lines violates the temporal constraints.

temporal constraints on neighborhood aggregation methods. **Thirdly**, node features and topological structures can exhibit temporal patterns. For example, node interactions that took place long ago may have less impact on the current topological structure and thus the node embeddings. Also, some nodes may possess features that allows them having more regular or recurrent interactions with others. We provide sketched plots for visual illustration in Figure 1.

Similar to its non-temporal counterparts, in the real-world applications, models for representation learning on temporal graphs should be able to quickly generate embeddings whenever required, in an *inductive* fashion. *GraphSAGE* (Hamilton et al., 2017a) and *graph attention network* (*GAT*) (Veličković et al., 2017) are capable of inductively generating embeddings for unseen nodes based on their features, however, they do not consider the temporal factors. Most of the temporal graph embedding methods can only handle *transductive* tasks, since they require re-training or the computationally-expensive gradient calculations to infer embeddings for unseen nodes or node embeddings for a new timepoint. In this work, we aim at developing an architecture to inductively learn representations for temporal graphs such that the time-aware embeddings (for unseen and observed nodes) can be obtained via a single network forward pass. The key to our approach is the combination of the self-attention mechanism (Vaswani et al., 2017) and a novel functional time encoding technique derived from the *Bochner's* theorem from classical harmonic analysis (Loomis, 2013).

The motivation for adapting self-attention to inductive representation learning on temporal graphs is to identify and capture relevant pieces of the temporal neighborhood information. Both graph convolutional network (*GCN*) (Kipf & Welling, 2016a) and *GAT* are implicitly or explicitly assigning different weights to neighboring nodes (Veličković et al., 2017) when aggregating node features. The self-attention mechanism was initially designed to recognize the relevant parts of input sequence in natural language processing. As a discrete-event sequence learning method, self-attention outputs a vector representation of the input sequence as a weighted sum of individual entry embeddings. Self-attention enjoys several advantages such as parallelized computation and interpretability (Vaswani et al., 2017). Since it captures sequential information only through the positional encoding, temporal features can not be handled. Therefore, we are motivated to replace positional encoding with some vector representation of time. Since time is a continuous variable, the mapping from the time domain to vector space has to be functional. We gain insights from harmonic analysis and propose a theoretical-grounded functional time encoding approach that is compatible with the self-attention mechanism. The temporal signals are then modelled by the interactions between the functional time encoding and nodes features as well as the graph topological structures.

To evaluate our approach, we consider future link prediction on the observed nodes as *transductive* learning task, and on the unseen nodes as *inductive* learning task. We also examine the dynamic node classification task using node embeddings (temporal versus non-temporal) as features to demonstrate the usefulness of our functional time encoding. We carry out extensive ablation studies and sensitivity analysis to show the effectiveness of the proposed functional time encoding and *TGAT*-layer.

## 2 RELATED WORK

**Graph representation learning**. Spectral graph embedding models operate on the graph spectral domain by approximating, projecting or expanding the graph Laplacian (Kipf & Welling, 2016a; Henaff et al., 2015; Defferrard et al., 2016). Since their training and inference are conditioned on the specific graph spectrum, they are not directly extendable to temporal graphs. Non-spectral approaches, such as *GAT*, *GraphSAGE* and *MoNET*, (Monti et al., 2017) rely on the localized neighbourhood aggregations and thus are not restricted to the training graph. *GraphSAGE* and *GAT* also have the flexibility to handle evolving graphs inductively. To extend classical graph representation learning approaches to the temporal domain, several attempts have been done by cropping the temporal graph into a sequence of graph snapshots (Li et al., 2018; Goyal et al., 2018; Rahman et al., 2018; Xu et al., 2019b), and some others work with temporally persistent node (edges) (Trivedi et al., 2018; Ma et al., 2018). Nguyen et al. (2018) proposes a node embedding method based on temporal random walk and reported state-of-the-art performances. However, their approach only generates embeddings for the final state of temporal graph and can not directly apply to the inductive setting.

**Self-attention mechanism.** Self-attention mechanisms often have two components: the embedding layer and the attention layer. The embedding layer takes an ordered entity sequence as input. Self-attention uses the positional encoding, i.e. each position $k$ is equipped with a vector $\mathbf{p}_k$ (fixed or learnt) which is shared for all sequences. For the entity sequence $\mathbf{e} = (e_1, \ldots, e_l)$, the embedding layer takes the sum or concatenation of entity embeddings (or features) ($\mathbf{z} \in \mathbb{R}^d$) and their positional encodings as input:

$$\mathbf{Z_e} = \left[ \mathbf{z}_{e_1} + \mathbf{p}_1, \ldots, \mathbf{z}_{e_1} + \mathbf{p}_l \right]^\intercal \in \mathbb{R}^{l \times d}, \text{ or } \quad \mathbf{Z_e} = \left[ \mathbf{z}_{e_1} || \mathbf{p}_1, \ldots, \mathbf{z}_{e_1} || \mathbf{p}_l \right]^\intercal \in \mathbb{R}^{l \times (d + d_{\text{pos}})}. \quad (1)$$

where $||$ denotes concatenation operation and $d_{\text{pos}}$ is the dimension for positional encoding. Self-attention layers can be constructed using the scaled dot-product attention, which is defined as:

$$\text{Attn}(\mathbf{Q}, \mathbf{K}, \mathbf{V}) = \text{softmax}\left( \frac{\mathbf{Q}\mathbf{K}^\intercal}{\sqrt{d}} \right) \mathbf{V}, \quad (2)$$

where $\mathbf{Q}$ denotes the 'queries', $\mathbf{K}$ the 'keys' and $\mathbf{V}$ the 'values'. In Vaswani et al. (2017), they are treated as projections of the output $\mathbf{Z_e}$: $\mathbf{Q} = \mathbf{Z_e}\mathbf{W}_Q, \quad \mathbf{K} = \mathbf{Z_e}\mathbf{W}_K, \quad \mathbf{V} = \mathbf{Z_e}\mathbf{W}_V$, where $\mathbf{W}_Q$, $\mathbf{W}_K$ and $\mathbf{W}_V$ are the projection matrices. Since each row of $\mathbf{Q}$, $\mathbf{K}$ and $\mathbf{V}$ represents an entity, the dot-product attention takes a weighted sum of the entity 'values' in $\mathbf{V}$ where the weights are given by the interactions of entity 'query-key' pairs. The hidden representation for the entity sequence under the dot-product attention is then given by $h_\mathbf{e} = \text{Attn}(\mathbf{Q}, \mathbf{K}, \mathbf{V})$.

## 3 TEMPORAL GRAPH ATTENTION NETWORK ARCHITECTURE

We first derive the mapping from time domain to the continuous differentiable functional domain as the functional time encoding such that resulting formulation is compatible with self-attention mechanism as well as the backpropagation-based optimization frameworks. The same idea was explored in a concurrent work (Xu et al., 2019a). We then present the temporal graph attention layer and show how it can be naturally extended to incorporate the edge features.

### 3.1 FUNCTIONAL TIME ENCODING

Recall that our starting point is to obtain a continuous functional mapping $\Phi : T \to \mathbb{R}^{d_T}$ from time domain to the $d_T$-dimensional vector space to replace the positional encoding in (1). Without loss of generality, we assume that the time domain can be represented by the interval starting from origin: $T = [0, t_{\max}]$, where $t_{\max}$ is determined by the observed data. For the inner-product self-attention in (2), often the 'key' and 'query' matrices ($\mathbf{K}$, $\mathbf{Q}$) are given by identity or linear projection of $\mathbf{Z_e}$ defined in (1), leading to terms that only involve inner-products between positional (time) encodings. Consider two time points $t_1, t_2$ and inner product between their functional encodings $\langle \Phi(t_1), \Phi(t_2) \rangle$. Usually, the relative timespan, rather than the absolute value of time, reveals critical temporal information. Therefore, we are more interested in learning patterns related to the timespan of $|t_2 - t_1|$, which should be ideally expressed by $\langle \Phi(t_1), \Phi(t_2) \rangle$ to be compatible with self-attention.

Formally, we define the temporal kernel $\mathcal{K} : T \times T \to \mathbb{R}$ with $\mathcal{K}(t_1, t_2) := \langle \Phi(t_1), \Phi(t_2) \rangle$ and $\mathcal{K}(t_1, t_2) = \psi(t_1 - t_2), \forall t_1, t_2 \in T$ for some $\psi : [-t_{\max}, t_{\max}] \to \mathbb{R}$. The temporal kernel is then

translation-invariant, since $\mathcal{K}(t_1 + c, t_2 + c) = \psi(t_1 - t_2) = \mathcal{K}(t_1, t_2)$ for any constant $c$. Generally speaking, functional learning is extremely complicated since it operates on infinite-dimensional spaces, but now we have transformed the problem into learning the temporal kernel $\mathcal{K}$ expressed by $\Phi$. Nonetheless, we still need to figure out an explicit parameterization for $\Phi$ in order to conduct efficient gradient-based optimization. Classical harmonic analysis theory, i.e. the Bochner's theorem, motivates our final solution. We point out that the temporal kernel $\mathcal{K}$ is positive-semidefinite (PSD) and continuous, since it is defined via Gram matrix and the mapping $\Phi$ is continuous. Therefore, the kernel $\mathcal{K}$ defined above satisfy the assumptions of the Bochner's theorem, which we state below.

**Theorem 1** (Bochner's Theorem). *A continuous, translation-invariant kernel $\mathcal{K}(\mathbf{x}, \mathbf{y}) = \psi(\mathbf{x} - \mathbf{y})$ on $\mathbb{R}^d$ is positive definite if and only if there exists a non-negative measure on $\mathbb{R}$ such that $\psi$ is the Fourier transform of the measure.*

Consequently, when scaled properly, our temporal kernel $\mathcal{K}$ have the alternate expression:

$$\mathcal{K}(t_1, t_2) = \psi(t_1, t_2) = \int_{\mathbb{R}} e^{i\omega(t_1 - t_2)} p(\omega) d\omega = \mathbb{E}_\omega[\xi_\omega(t_1)\xi_\omega(t_2)^*], \tag{3}$$

where $\xi_\omega(t) = e^{i\omega t}$. Since the kernel $\mathcal{K}$ and the probability measure $p(\omega)$ are real, we extract the real part of (3) and obtain:

$$\mathcal{K}(t_1, t_2) = \mathbb{E}_\omega\big[\cos(\omega(t_1 - t_2))\big] = \mathbb{E}_\omega\big[\cos(\omega t_1)\cos(\omega t_2) + \sin(\omega t_1)\sin(\omega t_2)\big]. \tag{4}$$

The above formulation suggests approximating the expectation by the Monte Carlo integral (Rahimi & Recht, 2008), i.e. $\mathcal{K}(t_1, t_2) \approx \frac{1}{d}\sum_{i=1}^{d}\cos(\omega_i t_1)\cos(\omega_i t_2) + \sin(\omega_i t_1)\sin(\omega_i t_2)$, with $\omega_1, \ldots, \omega_d \overset{\text{i.i.d}}{\sim} p(\omega)$. Therefore, we propose the finite dimensional functional mapping to $\mathbb{R}^d$ as:

$$t \mapsto \Phi_d(t) := \sqrt{\frac{1}{d}}\big[\cos(\omega_1 t), \sin(\omega_1 t), \ldots, \cos(\omega_d t), \sin(\omega_d t)\big], \tag{5}$$

and it is easy to show that $\langle \Phi_d(t_1), \Phi_d(t_2) \rangle \approx \mathcal{K}(t_1, t_2)$. As a matter of fact, we prove the stochastic uniform convergence of $\langle \Phi_d(t_1), \Phi_d(t_2) \rangle$ to the underlying $\mathcal{K}(t_1, t_2)$ and shows that it takes only a reasonable amount of samples to achieve proper estimation, which is stated in Claim 1.

**Claim 1.** *Let $p(\omega)$ be the corresponding probability measure stated in Bochner's Theorem for kernel function $\mathcal{K}$. Suppose the feature map $\Phi$ is constructed as described above using samples $\{\omega_i\}_{i=1}^{d}$, then we only need $d = \Omega\big(\frac{1}{\epsilon^2}\log\frac{\sigma_p^2 t_{\max}}{\epsilon}\big)$ samples to have*

$$\sup_{t_1, t_2 \in T}\big|\Phi_d(t_1)'\Phi_d(t_2) - \mathcal{K}(t_1, t_2)\big| < \epsilon \text{ with any probability for } \forall \epsilon > 0,$$

*where $\sigma_p^2$ is the second momentum with respect to $p(\omega)$.*

The proof is provided in supplement material.

By applying Bochner's theorem, we convert the problem of kernel learning to distribution learning, i.e. estimating the $p(\omega)$ in Theorem 1. A straightforward solution is to apply the *reparameterization* trick by using auxiliary random variables with a known marginal distribution as in variational autoencoders (Kingma & Welling, 2013). However, the *reparameterization* trick is often limited to certain distributions such as the 'local-scale' family, which may not be rich enough for our purpose. For instance, when $p(\omega)$ is multimodal it is difficult to reconstruct the underlying distribution via direct reparameterizations. An alternate approach is to use the inverse cumulative distribution function (CDF) transformation. Rezende & Mohamed (2015) propose using parameterized *normalizing flow*, i.e. a sequence of invertible transformation functions, to approximate arbitrarily complicated CDF and efficiently sample from it. Dinh et al. (2016) further considers stacking bijective transformations, known as affine coupling layer, to achieve more effective CDF estimation. The above methods learns the inverse CDF function $F_\theta^{-1}(.)$ parameterized by flow-based networks and draw samples from the corresponding distribution. On the other hand, if we consider an non-parameterized approach for estimating distribution, then learning $F^{-1}(.)$ and obtain $d$ samples from it is equivalent to directly optimizing the $\{\omega_1, \ldots, \omega_d\}$ in (4) as free model parameters. In practice, we find these two approaches to have highly comparable performances (see supplement material). Therefore we focus on the non-parametric approach, since it is more parameter-efficient and has faster training speed (as no sampling during training is required).

The above functional time encoding is fully compatible with self-attention, thus they can replace the positional encodings in (1) and their parameters are jointly optimized as part of the whole model.

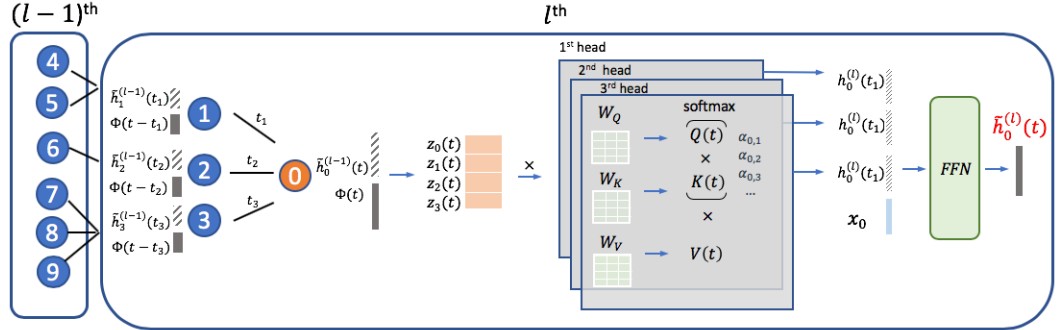

Figure 2: The architecture of the $l^{th}$ *TGAT* layer with $k = 3$ attention heads for node $v_0$ at time $t$.

## 3.2 TEMPORAL GRAPH ATTENTION LAYER

We use $v_i$ and $\mathbf{x}_i \in \mathbb{R}^{d_0}$ to denote node $i$ and its raw node features. The proposed TGAT architecture depends solely on the temporal graph attention layer (*TGAT* layer). In analogy to *GraphSAGE* and *GAT*, the *TGAT* layer can be thought of as a local aggregation operator that takes the temporal neighborhood with their hidden representations (or features) as well as timestamps as input, and the output is the time-aware representation for target node at any time point $t$. We denote the hidden representation output for node $i$ at time $t$ from the $l^{th}$ layer as $\tilde{\mathbf{h}}_i^{(l)}(t)$.

Similar to *GAT*, we perform the *masked self-attention* to take account of the structural information (Veličković et al., 2017). For node $v_0$ at time $t$, we consider its neighborhood $\mathcal{N}(v_0; t) = \{v_1, \ldots, v_N\}$ such that the interaction between $v_0$ and $v_i \in \mathcal{N}(v_0; t)$, which takes place at time $t_i$, is prior to $t$[1]. The input of *TGAT* layer is the neighborhood information $\mathbf{Z} = \{\tilde{\mathbf{h}}_1^{(l-1)}(t_1), \ldots, \tilde{\mathbf{h}}_N^{(l-1)}(t_N)\}$ and the target node information with some time point $(\tilde{\mathbf{h}}_0^{(l-1)}(t), t)$. When $l = 1$, i.e. for the first layer, the inputs are just raw node features. The layer produces the time-aware representation of target node $v_0$ at time $t$, denoted by $\tilde{\mathbf{h}}_0^{(l)}(t)$, as its output. Due to the translation-invariant assumption for the temporal kernel, we can alternatively use $\{t-t_1, \ldots, t-t_N\}$ as interaction times, since $|t_i - t_j| = |(t - t_i) - (t - t_j)|$ and we only care for the timespan.

In line with original self-attention mechanism, we first obtain the entity-temporal feature matrix as

$$\mathbf{Z}(t) = \left[\tilde{\mathbf{h}}_0^{(l-1)}(t)||\Phi_{d_T}(0), \tilde{\mathbf{h}}_1^{(l-1)}(t_1)||\Phi_{d_T}(t - t_1), \ldots, \tilde{\mathbf{h}}_N^{(l-1)}(t_N)||\Phi_{d_T}(t - t_N)\right]^{\mathsf{T}} \text{ (or use sum)}$$
(6)

and forward it to three different linear projections to obtain the 'query', 'key' and 'value':

$$\mathbf{q}(t) = [\mathbf{Z}(t)]_0 \mathbf{W}_Q, \, \mathbf{K}(t) = [\mathbf{Z}(t)]_{1:N} \mathbf{W}_K, \, \mathbf{V}(t) = [\mathbf{Z}(t)]_{1:N} \mathbf{W}_V,$$

where $\mathbf{W}_Q, \mathbf{W}_K, \mathbf{W}_V \in \mathbb{R}^{(d+d_T) \times d_h}$ are the weight matrices that are employed to capture the interactions between time encoding and node features. For notation simplicity, in the following discussion we treat the dependence of the intermediate outputs on target time $t$ as implicit. The attention weights $\{\alpha_i\}_{i=1}^N$ of the softmax function output in (2) is given by: $\alpha_i = \exp(\mathbf{q}^{\mathsf{T}}\mathbf{K}_i)/\left(\sum_q \exp(\mathbf{q}^{\mathsf{T}}\mathbf{K}_q)\right)$. The attention weight $\alpha_i$ reveals how node $i$ attends to the features of node $v_0$ within the topological structure defined as $\mathcal{N}(v_0; t)$ after accounting for their interaction time with $v_0$. The self-attention therefore captures the temporal interactions with both node features and topological features and defines a local temporal aggregation operator on graph. The hidden representation for any node $v_i \in \mathcal{N}(v_0; t)$ is given by: $\alpha_i \mathbf{V}_i$. The mechanism can be effectively shared across all nodes for any time point. We then take the row-wise sum from the above dot-product self-attention output as the hidden *neighborhood representations*, i.e.

$$\mathbf{h}(t) = \text{Attn}(\mathbf{q}(t), \mathbf{K}(t), \mathbf{V}(t)) \in \mathbb{R}^{d_h}.$$

---

[1] Node $v_i$ may have multiple interactions with $v_0$ at different time points. For the sake of presentation clarity, we do not explicitly differentiate such recurring interactions in our notations.

To combine neighbourhood representation with the target node features, we adopt the same practice from *GraphSAGE* and concatenate the neighbourhood representation with the target node's feature vector $\mathbf{z}_0$. We then pass it to a feed-forward neural network to capture non-linear interactions between the features as in (Vaswani et al., 2017):

$$\tilde{\mathbf{h}}_0^{(l)}(t) = \text{FFN}\Big(\mathbf{h}(t)||\mathbf{x}_0\Big) \equiv \text{ReLU}\Big([\mathbf{h}(t)||\mathbf{x}_0]\mathbf{W}_0^{(l)} + \mathbf{b}_0^{(l)}\Big)\mathbf{W}_1^{(l)} + \mathbf{b}_1^{(l)},$$

$$\mathbf{W}_0^{(l)} \in \mathbb{R}^{(d_h+d_0)\times d_f}, \mathbf{W}_1^{(l)} \in \mathbb{R}^{d_f \times d}, \mathbf{b}_0^{(l)} \in \mathbb{R}^{d_f}, \mathbf{b}_1^{(l)} \in \mathbb{R}^d,$$

where $\tilde{\mathbf{h}}_0^{(l)}(t) \in \mathbb{R}^d$ is the final output representing the time-aware node embedding at time $t$ for the target node. Therefore, the *TGAT* layer can be implemented for node classification task using the semi-supervised learning framework proposed in Kipf & Welling (2016a) as well as the link prediction task with the encoder-decoder framework summarized by Hamilton et al. (2017b).

Veličković et al. (2017) suggests that using *multi-head* attention improves performances and stabilizes training for *GAT*. For generalization purposes, we also show that the proposed TGAT layer can be easily extended to the *multi-head* setting. Consider the dot-product self-attention outputs from a total of $k$ different heads, i.e. $\mathbf{h}^{(i)} \equiv \text{Attn}^{(i)}\big(\mathbf{q}(t), \mathbf{K}(t), \mathbf{V}(t)\big)$, $i = 1, \ldots, k$. We first concatenate the $k$ neighborhood representations into a combined vector and then carry out the same procedure:

$$\tilde{\mathbf{h}}_0^{(l)}(t) = \text{FFN}\Big(\mathbf{h}^{(1)}(t)||\ldots||\mathbf{h}^{(k)}(t)||\mathbf{x}_0\Big).$$

Just like *GraphSAGE*, a single *TGAT* layer aggregates the localized one-hop neighborhood, and by stacking $L$ *TGAT* layers the aggregation extends to $L$ hops. Similar to *GAT*, out approach does not restrict the size of neighborhood. We provide a graphical illustration of our *TGAT* layer in Figure 2.

## 3.3 EXTENSION TO INCORPORATE EDGE FEATURES

We show that the *TGAT* layer can be naturally extended to handle edge features in a *message-passing* fashion. Simonovsky & Komodakis (2017) and Wang et al. (2018b) modify classical spectral-based graph convolutional networks to incorporate edge features. Battaglia et al. (2018) propose general graph neural network frameworks where edges features can be processed. For temporal graphs, we consider the general setting where each dynamic edge is associated with a feature vector, i.e. the interaction between $v_i$ and $v_j$ at time $t$ induces the feature vector $\mathbf{x}_{i,j}(t)$. To propagate edge features during the *TGAT* aggregation, we simply extend the $\mathbf{Z}(t)$ in (6) to:

$$\mathbf{Z}(t) = \Big[\ldots, \tilde{\mathbf{h}}_i^{(l-1)}(t_i)||\mathbf{x}_{0,i}(t_i)||\Phi_{d_T}(t - t_i), \ldots\Big] \text{ (or use summation)}, \tag{7}$$

such that the edge information is propagated to the target node's hidden representation, and then passed on to the next layer (if exists). The remaining structures stay the same as in Section 3.2.

## 3.4 TEMPORAL SUB-GRAPH BATCHING

Stacking $L$ *TGAT* layers is equivalent to aggregate over the $L$-hop neighborhood. For each $L$-hop sub-graph that is constructed during the batch-wise training, all message passing directions must be aligned with the observed chronological orders. Unlike the non-temporal setting where each edge appears only once, in temporal graphs two node can have multiple interactions at different time points. Whether or not to allow loops that involve the target node should be judged case-by-case. Sampling from neighborhood, or known as *neighborhood dropout*, may speed up and stabilize model training. For temporal graphs, neighborhood dropout can be carried uniformly or weighted by the inverse timespan such that more recent interactions has higher probability of being sampled.

## 3.5 COMPARISONS TO RELATED WORK

The functional time encoding technique and *TGAT* layer introduced in Section 3.1 and 3.2 solves several critical challenges, and the *TGAT* network intrinsically connects to several prior methods.

- Instead of cropping temporal graphs into a sequence of snapshots or constructing time-constraint random walks, which inspired most of the current temporal graph embedding methods, we directly learn the functional representation of time. The proposed approach is

     motivated by and thus fully compatible with the well-established self-attention mechanism. Also, to the best of our knowledge, no previous work has discussed the temporal-feature interactions for temporal graphs, which is also considered in our approach.

- The *TGAT* layer is computationally efficient compared with RNN-based models, since the masked self-attention operation is parallelizable, as suggested by Vaswani et al. (2017). The per-batch time complexity of the *TGAT* layer with $k$ heads and $l$ layers can be expressed as $O\big((k\tilde{N})^l\big)$ where $\tilde{N}$ is the average neighborhood size, which is comparable to *GAT*. When using *multi-head* attention, the computation for each head can be parallelized as well.

- The inference with *TGAT* is entirely *inductive*. With an explicit functional expression $\tilde{h}(t)$ for each node, the time-aware node embeddings can be easily inferred for any timestamp via a single network forward pass. Similarity, whenever the graph is updated, the embeddings for both unseen and observed nodes can be quickly inferred in an inductive fashion similar to that of *GraphSAGE*, and the computations can be parallelized across all nodes.

- *GraphSAGE* with mean pooling (Hamilton et al., 2017a) can be interpreted as a special case of the proposed method, where the temporal neighborhood is aggregated with equal attention coefficients. *GAT* is like the time-agnostic version of our approach but with a different formulation for self-attention, as they refer to the work of Bahdanau et al. (2014). We discuss the differences in detail in the Appendix. It is also straightforward to show our connections with the menory networks (Sukhbaatar et al., 2015) by thinking of the temporal neighborhoods as memory. The techniques developed in our work may also help adapting *GAT* and *GraphSAGE* to temporal settings as we show in our experiments.

## 4 EXPERIMENT AND RESULTS

We test the performance of the proposed method against a variety of strong baselines (adapted for temporal settings when possible) and competing approaches, for both the *inductive* and *transductive* tasks on two benchmark and one large-scale industrial dataset.

### 4.1 DATASETS

Real-world temporal graphs consist of time-sensitive node interactions, evolving node labels as well as new nodes and edges. We choose the following datasets which contain all scenarios.

**Reddit dataset**.[2] We use the data from active users and their posts under subreddits, leading to a temporal graph with 11,000 nodes, ∼700,000 temporal edges and dynamic labels indicating whether a user is banned from posting. The user posts are transformed into edge feature vectors.

**Wikipedia dataset**.[3] We use the data from top edited pages and active users, yielding a temporal graph ∼9,300 nodes and around 160,000 temporal edges. Dynamic labels indicate if users are temporarily banned from editing. The user edits are also treated as edge features.

**Industrial dataset**. We choose 70,000 popular products and 100,000 active customers as nodes from the online grocery shopping website[4] and use the customer-product purchase as temporal edges (∼2 million). The customers are tagged with labels indicating if they have a recent interest in dietary products. Product features are given by the pre-trained product embeddings (Xu et al., 2020).

We do the chronological train-validation-test split with 70%-15%-15% according to node interaction timestamps. The dataset and preprocessing details are provided in the supplement material.

### 4.2 TRANSDUCTIVE AND INDUCTIVE LEARNING TASKS

Since the majority of temporal information is reflected via the timely interactions among nodes, we choose to use a more revealing link prediction setup for training. Node classification is then treated as the downstream task using the obtained time-aware node embeddings as input.

---

[2]http://snap.stanford.edu/jodie/reddit.csv

[3]http://snap.stanford.edu/jodie/wikipedia.csv

[4]https://grocery.walmart.com/

| Dataset | Reddit | | Wikipedia | | Industrial | |
|---------|----------|------|----------|------|----------|------|
| Metric | Accuracy | AP | Accuracy | AP | Accuracy | AP |
| GAE | 74.31 (0.5) | 93.23 (0.3) | 72.85 (0.7) | 91.44 (0.1) | 68.92 (0.3) | 81.15 (0.2) |
| VAGE | 74.19 (0.4) | 92.92 (0.2) | 78.01 (0.3) | 91.34 (0.3) | 67.81 (0.4) | 80.87 (0.3) |
| DeepWalk | 71.43 (0.6) | 83.10 (0.5) | 76.67 (0.5) | 90.71 (0.6) | 65.87 (0.3) | 80.93 (0.2) |
| Node2vec | 72.53 (0.4) | 84.58 (0.5) | 78.09 (0.4) | 91.48 (0.3) | 66.64 (0.3) | 81.39 (0.3) |
| CTDNE | 73.76 (0.5) | 91.41 (0.3) | 79.42 (0.4) | 92.17 (0.5) | 67.81 (0.3) | 80.95 (0.5) |
| GAT | 92.14 (0.2) | 97.33 (0.2) | 87.34 (0.3) | 94.73 (0.2) | 69.58 (0.4) | 81.51 (0.2) |
| GAT+T | 92.47 (0.2) | 97.62 (0.2) | 87.57 (0.2) | 95.14 (0.4) | 70.15 (0.3) | 82.66 (0.4) |
| GraphSAGE | 92.31(0.2) | 97.65 (0.2) | 85.93 (0.3) | 93.56 (0.3) | 70.19 (0.2) | 83.27 (0.3) |
| GraphSAGE+T | 92.58 (0.2) | 97.89 (0.3) | 86.31 (0.3) | 93.72 (0.3) | 71.84 (0.3) | 84.95 (0.) |
| Const-TGAT | 91.39 (0.2) | 97.86 (0.2) | 86.03 (0.4) | 93.50 (0.3) | 68.52 (0.2) | 81.91 (0.3) |
| TGAT | **92.92** (0.3) | **98.12** (0.2) | **88.14** (0.2) | **95.34** (0.1) | **73.28** (0.2) | **86.32** (0.1) |

Table 1: Transductive learning task results for predicting future edges of nodes that have been observed in training data. All results are converted to percentage by multiplying by 100, and the standard deviations computed over ten runs (in parenthesis). The best and second-best results in each column are highlighted in **bold** font and underlined. *GraphSAGE* is short for *GraphSAGE*-LSTM.

| Dataset | Reddit | | Wikipedia | | Industrial | |
|---------|----------|------|----------|------|----------|------|
| Metric | Accuracy | AP | Accuracy | AP | Accuracy | AP |
| GAT | 89.86 (0.2) | 95.37 (0.3) | 82.36 (0.3) | 91.27 (0.4) | 68.28 (0.2) | 79.93 (0.3) |
| GAT+T | 90.44 (0.3) | 96.31 (0.3) | 84.82 (0.3) | 93.57 (0.3) | 69.51 (0.3) | 81.68 (0.3) |
| GraphSAGE | 89.43 (0.1) | 96.27 (0.2) | 82.43 (0.3) | 91.09 (0.3) | 67.49 (0.2) | 80.54 (0.3) |
| GraphSAGE+T | 90.07 (0.2) | 95.83 (0.2) | 84.03 (0.4) | 92.37 (0.5) | 69.66 (0.3) | 82.74 (0.3) |
| Const-TGAT | 88.28 (0.3) | 94.12 (0.2) | 83.60 (0.4) | 91.93 (0.3) | 65.87 (0.3) | 77.03 (0.4) |
| TGAT | **90.73** (0.2) | **96.62** (0.3) | **85.35** (0.2) | **93.99** (0.3) | **72.08** (0.3) | **84.99** (0.2) |

Table 2: Inductive learning task results for predicting future edges of unseen nodes.

**Transductive task** examines embeddings of the nodes that have been observed in training, via the future link prediction task and the node classification. To avoid violating temporal constraints, we predict the links that strictly take place posterior to all observations in the training data.

**Inductive task** examines the *inductive* learning capability using the inferred representations of unseen nodes, by predicting the future links between unseen nodes and classify them based on their inferred embedding dynamically. We point out that it suffices to only consider the future sub-graph for unseen nodes since they are equivalent to new graphs under the non-temporal setting.

As for the evaluation **metrics**, in the link prediction tasks, we first sample an equal amount of negative node pairs to the positive links and then compute the *average precision* (*AP*) and classification *accuracy*. In the downstream node classification tasks, due to the label imbalance in the datasets, we employ the *area under the ROC curve* (*AUC*).

## 4.3 BASELINES

**Transductive task**: for link prediction of observed nodes, we choose the compare our approach with the state-of-the-art graph embedding methods: *GAE* and *VGAE* (Kipf & Welling, 2016b). For complete comparisons, we also include the skip-gram-based *node2vec* (Grover & Leskovec, 2016) as well as the spectral-based *DeepWalk* model (Perozzi et al., 2014), using the same inner-product decoder as *GAE* for link prediction. The *CDTNE* model based on the temporal random walk has been reported with superior performance on transductive learning tasks (Nguyen et al., 2018), so we include *CDTNE* as the representative for temporal graph embedding approaches.

**Inductive task**: few approaches are capable of managing inductive learning on graphs even in the non-temporal setting. As a consequence, we choose *GraphSAGE* and *GAT* as baselines after adapting them to the temporal setting. In particular, we equip them with the same *temporal sub-graph batching* describe in Section 3.4 to maximize their usage on temporal information. Also, we implement the extended version for the baselines to include edge features in the same way as ours (in Section 3.3). We experiment on different aggregation functions for *GraphSAGE*, i.e. *Graph-*

*SAGE*-mean, *GraphSAGE*-pool and *GraphSAGE*-LSTM. In accordance with the original work of Hamilton et al. (2017a), *GraphSAGE*-LSTM gives the best validation performance among the three approaches, which is reasonable under temporal setting since LSTM aggregation takes account of the sequential information. Therefore we report the results of *GraphSAGE*-LSTM.

In addition to the above baselines, we implement a version of *TGAT* with all temporal attention weights set to equal value (*Const-TGAT*). Finally, to show that the superiority of our approach owes to both the time encoding and the network architecture, we experiment with the enhanced *GAT* and *GraphSAGE*-mean by concatenating the proposed time encoding to the original features during temporal aggregations (*GAT+T* and *GraphSAGE+T*).

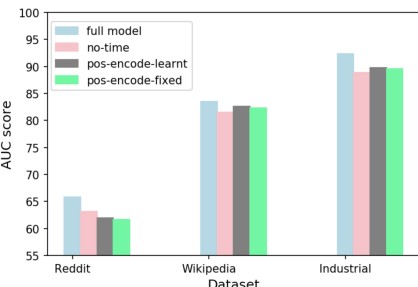

| Dataset | Reddit | Wikipedia | Industrial |
|---|---|---|---|
| GAE | 58.39 (0.5) | 74.85 (0.6) | 76.59 (0.3) |
| VGAE | 57.98 (0.6) | 73.67 (0.8) | 75.38 (0.4) |
| CTDNE | 59.43 (0.6) | 75.89 (0.5) | 78.36 (0.5) |
| GAT | 64.52 (0.5) | 82.34 (0.8) | 87.43 (0.4) |
| GAT+T | 64.76 (0.6) | 82.95 (0.7) | 88.24 (0.5) |
| GraphSAGE | 61.24 (0.6) | 82.42 (0.7) | 88.28 (0.3) |
| GraphSAGE+T | 62.31 (0.7) | 82.87 (0.6) | 89.81 (0.3) |
| Const-TGAT | 60.97 (0.5) | 75.18 (0.7) | 82.59 (0.6) |
| TGAT | **65.56** (0.7) | **83.69** (0.7) | **92.31** (0.3) |

Figure 3: Results of node classification task in the **ablation study**.

Table 3: Dynamic node classification task results, where the reported metric is the *AUC*.

### 4.4 EXPERIMENT SETUP

We use the time-sensitive link prediction loss function for training the *l*-layer *TGAT* network:

$$\ell = \sum_{(v_i, v_j, t_{ij}) \in \mathcal{E}} - \log \left( \sigma \left( - \tilde{\mathbf{h}}_i^l(t_{ij})^\mathsf{T} \tilde{\mathbf{h}}_j^l(t_{ij}) \right) \right) - Q.\mathbb{E}_{v_q \sim P_n(v)} \log \left( \sigma \big( \tilde{\mathbf{h}}_i^l(t_{ij})^\mathsf{T} \tilde{\mathbf{h}}_q^l(t_{ij}) \big) \right), \quad (8)$$

where the summation is over the observed edges on $v_i$ and $v_j$ that interact at time $t_{ij}$, and $\sigma(.)$ is the sigmoid function, $Q$ is the number of negative samples and $P_n(v)$ is the negative sampling distribution over the node space. As for tuning hyper-parameters, we fix the node embedding dimension and the time encoding dimension to be the original feature dimension for simplicity, and then select the number of *TGAT* layers from {1,2,3}, the number of attention heads from {1,2,3,4,5}, according to the link prediction *AP* score in the validation dataset. Although our method does not put restriction on the neighborhood size during aggregations, to speed up training, specially when using the multi-hop aggregations, we use neighborhood dropout (selected among $p =$ {0.1, 0.3, 0.5}) with the uniform sampling. During training, we use 0.0001 as learning rate for Reddit and Wikipedia dataset and 0.001 for the industrial dataset, with Glorot initialization and the Adam SGD optimizer. We do not experiment on applying regularization since our approach is parameter-efficient and only requires $\Omega\big((d + d_T)d_h + (d_h + d_0)d_f + d_f d\big)$ parameters for each attention head, which is independent of the graph and neighborhood size. Using two *TGAT* layers and two attention heads with dropout rate as 0.1 give the best validation performance. For inference, we *inductively* compute the embeddings for both the unseen and observed nodes at each time point that the graph evolves, or when the node labels are updated. We then use these embeddings as features for the future link prediction and dynamic node classifications with multilayer perceptron.

We further conduct **ablation study** to demonstrate the effectiveness of the proposed functional time encoding approach. We experiment on abandoning time encoding or replacing it with the original positional encoding (both fixed and learnt). We also compare the uniform neighborhood dropout to sampling with inverse timespan (where the recent edges are more likely to be sampled), which is provided in supplement material along with other implementation details and setups for baselines.

### 4.5 RESULTS

The results in Table 1 and Table 2 demonstrates the state-of-the-art performances of our approach on both *transductive* and *inductive* learning tasks. In the *inductive* learning task, our *TGAT* network significantly improves upon the the upgraded *GraphSAGE*-LSTM and *GAT* in *accuracy* and *average*

*precision* by at least 5 % for both metrics, and in the *transductive* learning task *TGAT* consistently outperforms all baselines across datasets. While *GAT+T* and *GraphSAGE+T* slightly outperform or tie with *GAT* and *GraphSAGE*-LSTM, they are nevertheless outperformed by our approach. On one hand, the results suggest that the time encoding have potential to extend non-temporal graph representation learning methods to temporal settings. On the other, we note that the time encoding still works the best with our network architecture which is designed for temporal graphs. Over-all, the results demonstrate the superiority of our approach in learning representations on temporal graphs over prior models. We also see the benefits from assigning temporal attention weights to neighboring nodes, where *GAT* significantly outperforms the *Const-TGAT* in all three tasks. The dynamic node classification outcome (in Table 3) further suggests the *usefulness* of our time-aware node embeddings for downstream tasks as they surpass all the baselines. The **ablation study** results of Figure 3 successfully reveals the effectiveness of the proposed functional time encoding approach in capturing temporal signals as it outperforms the positional encoding counterparts.

### 4.6 ATTENTION ANALYSIS

To shed some insights into the temporal signals captured by the proposed *TGAT*, we analyze the pattern of the attention weights $\{\alpha_{ij}(t)\}$ as functions of both time $t$ and node pairs $(i, j)$ in the inference stage. **Firstly**, we analyze how the attention weights change with respect to the timespans of previous interactions, by plotting the attention weights $\{\alpha_{jq}(t_{ij})|q \in \mathcal{N}(v_j; t_{ij})\} \cup \{\alpha_{ik}(t_{ij})|k \in \mathcal{N}(v_i; t_{ij})\}$ against the timespans $\{t_{ij}-t_{jq}\} \cup \{t_{ij}-t_{ik}\}$ when predicting the link for $(v_i, v_j, t_{ij}) \in \mathcal{E}$ (Figure 4a). This gives us an empirical estimation on the $\alpha(\Delta t)$, where a smaller $\Delta t$ means a more recent interaction. **Secondly**, we analyze how the topological structures affect the attention weights as time elapses. Specifically, we focus on the topological structure of the *recurring neighbours*, by finding out what attention weights the model put on the neighbouring nodes with different number of *reoccurrences*. Since the functional forms of all $\{\alpha_{ij}(.)\}$ are fixed after training, we are able to feed in different target time $t$ and then record their value on neighbouring nodes with different number of occurrences (Figure 4b). From Figure 4a we observe that *TGAT* captures the pattern of having less attention on more distant interactions in all three datasets. In Figure 4b, it is obvious that when predicting a more future interaction, *TGAT* will consider neighbouring nodes who have a higher number of occurrences of more importance. The patterns of the attention weights are meaningful, since the more recent and repeated actions often have larger influence on users' future interests.

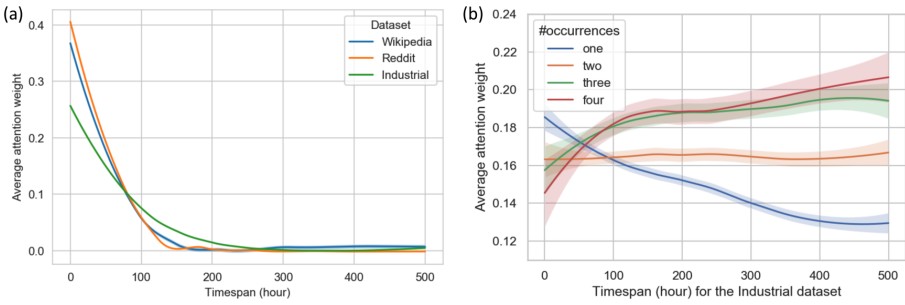

Figure 4: Attention weight analysis. We apply the *Loess smoothing* method for visualization.

## 5 CONCLUSION AND FUTURE WORK

We introduce a novel time-aware graph attention network for inductive representation learning on temporal graphs. We adapt the self-attention mechanism to handle the continuous time by proposing a theoretically-grounded functional time encoding. Theoretical and experimental analysis demon-strate the effectiveness of our approach for capturing temporal-feature signals in terms of both node and topological features on temporal graphs. Self-attention mechanism often provides useful model interpretations (Vaswani et al., 2017), which is an important direction of our future work. Develop-ing tools to visualize the evolving graph dynamics and temporal representations efficiently is another important direction for both research and application. Also, the functional time encoding technique has huge potential for adapting other deep learning methods to the temporal graph domain.

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

# A APPENDIX

## A.1 PROOF FOR CLAIM 1

*Proof.* The proof is also shown in our concurrent work Xu et al. (2019a). We also provide it here for completeness. To prove the results in Claim 1, we alternatively show that under the same condition,

$$\Pr\big(\sup_{t_1,t_2\in T} |\Phi_d^{\mathcal{B}}(t_1)^{'}\Phi_d^{\mathcal{B}}(t_2) - \mathcal{K}(t_1,t_2)| \geq \epsilon\big) \leq 4\sigma_p\sqrt{\frac{t_{\max}}{\epsilon}}exp\big(\frac{-d\epsilon^2}{32}\big). \tag{9}$$

Define the score $S(t_1,t_2) = \Phi_d^{\mathcal{B}}(t_1)^{'}\Phi_d^{\mathcal{B}}(t_2)$. The goal is to derive a uniform upper bound for $s(t_1,t_2) - \mathcal{K}(t_1,t_2)$. By assumption $S(t_1,t_2)$ is an unbiased estimator for $\mathcal{K}(t_1,t_2)$, i.e. $E[S(t_1,t_2)] = \mathcal{K}(t_1,t_2)$. Due to the translation-invariant property of $S$ and $\mathcal{K}$, we let $\Delta(t) \equiv s(t_1,t_2) - \mathcal{K}(t_1,t_2)$, where $t \equiv t_1 - t_2$ for all $t_1,t_2 \in [0,t_{\max}]$. Also we define $s(t_1 - t_2) := S(t_1,t_2)$. Therefore $t \in [-t_{\max},t_{\max}]$, and we use $t \in \tilde{T}$ as the shorthand notation. The LHS in (1) now becomes $\Pr\big(\sup_{t\in\tilde{T}} |\Delta(t)| \geq \epsilon\big)$.

Note that $\tilde{T} \subseteq \cup_{i=0}^{N-1}T_i$ with $T_i = [-t_{\max} + \frac{2it_{\max}}{N}, -t_{\max} + \frac{2(i+1)t_{\max}}{N}]$ for $i = 1,\ldots,N$. So $\cup_{i=0}^{N-1}T_i$ is a finite cover of $\tilde{T}$. Define $t_i = -t_{\max} + \frac{(2i+1)t_{\max}}{N}$, then for any $t \in T_i$, $i = 1,\ldots,N$ we have

$$\begin{aligned} |\Delta(t)| &= |\Delta(t) - \Delta(t_i) + \Delta(t_i)| \\ &\leq |\Delta(t) - \Delta(t_i)| + |\Delta(t_i)| \\ &\leq L_\Delta|t - t_i| + |\Delta(t_i)| \\ &\leq L_\Delta\frac{2t_{\max}}{N} + |\Delta(t_i)|, \end{aligned} \tag{10}$$

where $L_\Delta = \max_{t\in\tilde{T}} \|\nabla\Delta(t)\|$ (since $\Delta$ is differentiable) with the maximum achieved at $t^*$. So we may bound the two events separately.

For $|\Delta(t_i)|$ we simply notice that trigeometric functions are bounded between $[-1,1]$, and therefore $-1 \leq \Phi_d^{\mathcal{B}}(t_1)^{'}\Phi_d^{\mathcal{B}}(t_2) \leq 1$. The Hoeffding's inequality for bounded random variables immediately gives us:

$$\Pr\big(|\Delta(t_i)| > \frac{\epsilon}{2}\big) \leq 2exp(-\frac{d\epsilon^2}{16}).$$

So applying the Hoeffding-type union bound to the finite cover gives

$$\Pr(\cup_{i=0}^{N-1}|\Delta(t_i)| \geq \frac{\epsilon}{2}) \leq 2N\exp(-\frac{d\epsilon^2}{16}) \tag{11}$$

For the other event we first apply Markov inequality and obtain:

$$\Pr\big(L_\Delta\frac{2t_{\max}}{N} \geq \frac{\epsilon}{2}\big) = \Pr\big(L_\Delta \geq \frac{\epsilon N}{4t_{\max}}\big) \leq \frac{4t_{\max}E[L_\Delta^2]}{\epsilon N}. \tag{12}$$

Also, since $E[s(t_1 - t_2)] = \psi(t_1 - t_2)$, we have

$$E[L_\Delta^2] = E\|\nabla s(t^*) - \nabla\psi(t^*)\|^2 = E\|\nabla s(t^*)\|^2 - E\|\nabla\psi(t^*)\|^2 \leq E\|\nabla s(t^*)\|^2 = \sigma_p^2, \tag{13}$$

where $\sigma_p^2$ is the second momentum with respect to $p(\omega)$.

Combining (11), (12) and (11) gives us:

$$\Pr\big(\sup_{t\in\tilde{T}} |\Delta(t)| \geq \epsilon\big) \leq 2N\exp(-\frac{d\epsilon^2}{16}) + \frac{4t_{\max}\sigma_p^2}{\epsilon N}. \tag{14}$$

It is straightforward to examine that the RHS of (14) is a convex function of $N$ and is minimized by $N^* = \sigma_p\sqrt{\frac{2t_{\max}}{\epsilon}}exp(\frac{d\epsilon^2}{32})$. Plug $N^*$ back to (14) and we obtain (9). We then solve for $d$ according to (9) and obtain the results in Claim 1.

$$\square$$

## A.2 COMPARISONS BETWEEN THE ATTENTION MECHANISM OF *TGAT* AND *GAT*

In this part, we provide detailed comparisons between the attention mechanism employed by our proposed *TGAT* and the *GAT* proposed by Veličković et al. (2017). Other than the obvious fact that *GAT* does not handle temporal information, the main difference lies in the formulation of attention weights. While *GAT* depends on the attention mechanism proposed by Bahdanau et al. (2014), our architecture refers to the self-attention mechanism of Vaswani et al. (2017). Firstly, the attention mechanism used by *GAT* does not involve the notions of 'query', 'key' and 'value' nor the dot-product formulation introduced in (2). As a consequence, the attention weight between node $v_i$ and its neighbor $v_j$ is computed via

$$\alpha_{ij} = \frac{\exp\left(\text{LeakyReLU}\left(\mathbf{a}^\intercal[\mathbf{W}\mathbf{h}_i||\mathbf{W}\mathbf{h}_j]\right)\right)}{\sum_{k\in\mathcal{N}(v_i)}\exp\left(\text{LeakyReLU}\left(\mathbf{a}^\intercal[\mathbf{W}\mathbf{h}_i||\mathbf{W}\mathbf{h}_k]\right)\right)},$$

where $\mathbf{a}$ is a weight vector, $\mathbf{W}$ is a weight matrix, $\mathcal{N}(v_i)$ is the neighborhood set for node $v_i$ and $\mathbf{h}_i$ is the hidden representation of node $v_i$. It is then obvious that their computation of $\alpha_{ij}$ is very different from our approach. In *TGAT*, after expanding the expressions in Section 3, the attention weight is computed by:

$$\alpha_{ij}(t) = \frac{\exp\left(\left([\tilde{\mathbf{h}}_i(t_i)||\Phi_{d_T}(t-t_i)]\mathbf{W}_Q\right)^\intercal\left([\tilde{\mathbf{h}}_j(t_j)||\Phi_{d_T}(t-t_j)]\mathbf{W}_K\right)\right)}{\sum_{k\in\mathcal{N}(v_i;t)}\exp\left(\left([\tilde{\mathbf{h}}_i(t_i)||\Phi_{d_T}(t-t_i)]\mathbf{W}_Q\right)^\intercal\left([\tilde{\mathbf{h}}_k(t_k)||\Phi_{d_T}(t-t_k)]\mathbf{W}_K\right)\right)}.$$

Intuitively speaking, the attention mechanism of *GAT* relies on the parameter vector $\mathbf{a}$ and the LeakyReLU(.) to capture the hidden factor interactions between entities in the sequence, while we use the linear transformation followed by the dot-product to capture pair-wise interactions of the hidden factors between entities and the time embeddings. The dot-product formulation is important for our approach. From the theoretical perspective, the time encoding functional form is derived according to the notion of temporal kernel $\mathcal{K}$ and its inner-product decomposition (Section 3). As for the practical performances, we see from Table 1, 2 and 3 that even after we equip *GAT* with the same time encoding, the performance is still inferior to our *TGAT*.

## A.3 DETAILS ON DATASETS AND PREPROCESSING

**Reddit dataset**: this benchmark dataset contains users interacting with subreddits by posting under the subreddits. The timestamps tell us when the user makes the posts. The dataset uses the posts made in a one-month span, and selects the most active users and subreddits as nodes, giving a total of 11,000 nodes and around 700,000 temporal edges. The user posts have textual features that are transformed into a 172-dimensional vector representing under the *linguistic inquiry and word count* (LIWC) categories (Pennebaker et al., 2001). The dynamic binary labels indicate if a user is banned from posting under a subreddit. Since node features are not provided in the original dataset, we use the all-zero vector instead.

**Wikipedia dataset**: the dataset also collects one-month of interactions induced by users' editing the Wikipedia pages. The the top edited pages and active users are considered, leading to $\sim$9,300 nodes and around 160,000 temporal edges. Similar to the Reddit dataset, we also have the ground-truth dynamic labels on whether a user is banned from editing a Wikipedia page. User edits consist of the textual features and are also converted into 172-dimensional LIWC feature vectors. Node features are also not provided, so we also use the all-zero vector as well.

**Industrial dataset**: we obtain the large-scale customer-product interaction graph from the online grocery shopping platform `grocery.walmart.com`. We select $\sim$70,000 most popular products and 100,000 active customers as nodes and use the customer-product purchase interactions over a one-month period as temporal edges ($\sim$2 million). Each purchase interaction is timestamped, which we use to construct the temporal graph. The customers are labelled with business tags, indicating if they are interested in dietary products according to their most recent purchase records. Each product node possesses contextual features containing their name, brand, categories and short description. The previous LIWC categories no longer apply since the product contextual features are not natural sentences. We use product embedding approach (Xu et al., 2020) to embed each product's contextual

|  | Reddit | Wikipedia | Industrial |
|---|---|---|---|
| # Nodes | 11,000 | 9,227 | 170,243 |
| # Edges | 672,447 | 157,474 | 2,135,762 |
| # Feature dimension | 172 | 172 | 100 |
| # Feature type | LIWC category vector | LIWC category vector | document embeddings |
| # Timespan | 30 days | 30 days | 30 days |
| % Training nodes | 90% | 90% | 90% |
| % Unseen nodes | 10% | 10% | 10% |
| % Training edges | ∼67% | ∼65% | ∼64% |
| % Future edges between observed nodes | ∼27% | ∼28% | ∼29% |
| % Future edges between unseen nodes | ∼6% | ∼7% | ∼7% |
| # Nodes with dynamic labels | 366 | 217 | 5,236 |
| Label type | binary | binary | binary |
| Positive label meaning | banned from posting | banned from editting | interested in dietary products |

Table 4: Data statistics for the three datasets. Since we sample a proportion of unseen nodes, the percentage of the edge statistics reported here are approximations.

features into a 100-dimensional vector space as preprocessing. The user nodes and edges do not possess features.

We then split the temporal graphs chronologically into 70%-15%-15% for training, validation and testing according to the time epochs of edges, as illustrated in Figure 5 with the Reddit dataset. Since all three datasets have a relatively stationary edge count distribution over time, using the 70 and 85 percentile time points to split the dataset results in approximately 70%-15%-15% of total edges, as suggested by Figure 5.

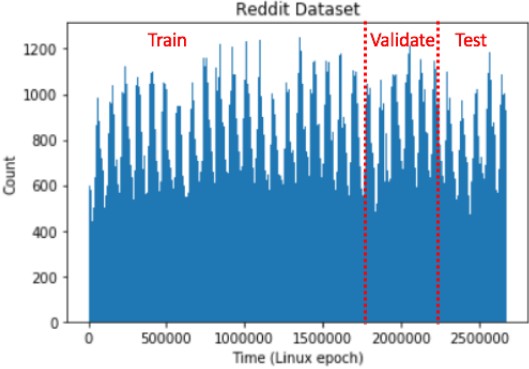

Figure 5: The distribution of temporal edge count for the Reddit dataset, and the illustration on the train-validation-test splitting.

To ensure that an appropriate amount of future edges among the unseen nodes will show up during validation and testing, for each dataset, we randomly sample 10% of nodes, mask them during training and treat them as unseen nodes by only considering their interactions in validation and testing period. This manipulation is necessary since the new nodes that show up during validation and testing period may not have much interaction among themselves. The statistics for the three datasets are summarized in Table 4.

**Preprocessing.**

For the *Node2vec* and *DeepWalk* baselines who only take static graphs as input, the graph is constructed using all edges in training data regardless of temporal information. For *DeepWalk*, we treat

the recurrent edges as appearing only once, so the graph is unweighted. Although our approach handles both directed and undirected graphs, for the sake of training stability of the baselines, we treat the graphs as *undirected*. For *Node2vec*, we use the count of recurrent edges as their weights and construct the weighted graph. For all three datasets, the obtained graphs in both cases are *undirected* and do not have isolated nodes. Since we choose from active users and popular items, the graphs are all *connected*.

For the graph convolutional network baselines, i.e. *GAE* and *VGAE*, we construct the same undirected weighted graph as for *Node2vec*. Since *GAE* and *VGAE* do not take edge features as input, we use the posts/edits as user node features. For each user in Reddit and Wikipedia dataset, we take the average of their post/edit feature vectors as the node feature. For the industrial dataset where user features are not available, we use the all-zero feature vector instead.

As for the downstream dynamic node classification task, we use the same training, validation and testing dataset as above. Since we aim at predicting the dynamic node labels, for Reddit and Wikipedia dataset we predict if the user node is banned and for the industrial dataset we predict the customers' business labels, at different time points. Due to the label imbalance, in each of the batch when training for the node label classifier, we conduct stratified sampling such that the label distributions are similar across batches.

## A.4 EXPERIMENT SETUP FOR BASELINES

For all baselines, we set the node embedding dimension to $d = 100$ to keep in accordance with our approach.

**Transductive baselines.**

Since *Node2vec* and *DeepWalk* do not provide room for task-specific manipulation or hacking, we do not modify their default loss function and input format. For both approaches, we select the *number of walks* among {60,80,100} and the *walk-length* among {20,30,40} according to the validation *AP*. Setting *number of walks*=80 and *walk-length*=30 give slightly better validation performance compared to others for both approaches. Notice that both *Node2vec* and *DeepWalk* use the sigmoid function with embedding inner-products as the decoder to predict neighborhood probabilities. So when predicting whether $v_i$ and $v_j$ will interact in the future, we use $\sigma(-\mathbf{z}_i^\mathsf{T}\mathbf{z}_j)$ as the score, where $\mathbf{z}_i$ and $\mathbf{z}_j$ are the node embeddings. Notice that *Node2vec* has the extra hyper-parameter $p$ and $q$ which controls the likelihood of immediately revisiting a node in the walk and interpolation between breadth-first strategy and depth-first strategy. After selecting the optimal *number of walks* and *walk-length* under $p = 1$ and $q = 1$, we further tune the different values of $p$ in {0.2,0.4,0.6,0.8,1.0} while fixing $q = 1$. According to validation, $p = 0.6$ and $0.8$ give comparable optimal performance.

For the *GAE* and *VGAE* baselines, we experiment on using one, two and three graph convolutional layers as the encoder (Kipf & Welling, 2016a) and use the ReLU(.) as the activation function. By referencing the official implementation, we also set the dimension of hidden layers to 200. Similar to previous findings, using two layers gives significant performances to using only one layer. Adding the third layer, on the other hand, shows almost identical results for both models. Therefore the results reported are based on two-layer *GCN* as the encoder. For *GAE*, we use the standard inner-product decoder as our approach and optimize over the reconstruction loss, and for *VGAE*, we restrict the Gaussian latent factor space (Kipf & Welling, 2016b). Since we have eliminated the temporal information when constructing the input, we find that the optimal hyper-parameters selected according to the tuning have similar patterns as in the previous non-temporal settings.

For the temporal network embedding model *CTDNE*, the *walk length* for the temporal random walk is also selected among {60,80,100}, where setting *walk length* to 80 gives slightly better validation outcome. The original paper considers several temporal edge selection (sampling) methods (uniform, linear and exponential) and finds uniform sampling with best performances (Nguyen et al., 2018). Since our setting is similar to theirs, we adopt the uniform sampling approach.

**Inductive baselines.**

For the *GraphSAGE* and *GAT* baselines, as mentioned before, we train the models in an identical way as our approach with the *temporal subgraph batching*, despite several slight differences. Firstly, the aggregation layers in *GraphSAGE* usually considers a fixed neighborhood size via sampling,

whereas our approach can take an arbitrary neighborhood as input. Therefore, we only consider the most recent $d_{\text{sample}}$ edges during each aggregation for all layers, and we find $d_{\text{sample}} = 20$ gives the best performance among $\{10,15,20,25\}$. Secondly, *GAT* implements a uniform neighborhood dropout. We also experiment with the inverse timespan sampling for neighborhood dropout, and find that it gives slightly better performances but at the cost of computational efficiency, especially for large graphs. We consider aggregating over one, two and three-hop neighborhood for both *GAT* and *GraphSAGE*. When working with three hops, we only experiment on *GraphSAGE* with the mean pooling aggregation. In general, using two hops gives comparable performance to using three hops. Notice that computations with three-hop are costly, since the number of edges during aggregation increase exponentially to the number of hops. Thus we stick to using two hops for *GraphSAGE*, *GAT* and our approach. It is worth mentioning that when implementing *GraphSAGE*-LSTM, the input neighborhood sequences of LSTM are also ordered by their interaction time.

**Node classification with baselines.**

The dynamic node classification with *GraphSAGE* and *GAT* can be conducted similarity to our approach, where we inductively compute the most up-to-date node embeddings and then input them as features to an MLP classifier. For the transductive baselines, it is not reasonable to predict the dynamic node labels with only the fixed node embeddings. Instead, we combine the node embedding with the other node embedding it is interacting with when the label changes, e.g. combine the user embedding with the Wikipedia page embedding that the user attempts on editing when the system bans the user. To combine the pair of node embeddings, we experimented on summation, concatenation and bi-linear transformation. Under summation and concatenation, the combined embeddings are then used as input to an MLP classifier, where the bi-linear transformation directly outputs scores for classification. The validation outcomes suggest that using concatenation with MLP yields the best performance.

## A.5 IMPLEMENTATION DETAILS

**Training.** We implement *Node2vec* using the official C code[5] on a 16-core Linux server with 500 Gb memory. *DeepWalk* is implemented with the official python code[6]. We refer to the PyTorch geometric library for implementing the *GAE* and *VGAE* baselines (Fey & Lenssen, 2019). To accommodate the temporal setting and incorporate edges features, we develop off-the-shelf implementation for *GraphSAGE* and *GAT* in PyTorch by referencing their original implementations[7] [8]. We also implement our model using PyTorch. All the deep learning models are trained on a machine with one Tesla V100 GPU. We use the Glorot initialization and the Adam SGD optimizer for all models, and apply the early-stopping strategy during training where we terminate the training process if the validation *AP* score does not improve for 10 epochs.

**Downstream node classification.** As we discussed before, we use the three-layer MLP as classifier and the (combined) node embeddings as input features from all the experimented approaches, for all three datasets. The MLP is trained with the Glorot initialization and the Adam SGD optimizer in PyTorch as well. The $\ell_2$ regularization parameter $\lambda$ is selected in $\{0.001, 0.01, 0.05, 0.1, 0.2\}$ case-by-case during training. The early-stopping strategy is also employed.

## A.6 SENSITIVITY ANALYSIS AND EXTRA ABLATION STUDY

Firstly, we focus on the output node embedding dimension as well as the functional time encoding dimension in this sensitivity analysis. The reported results are averaged over five runs. We experiment on $d \in \{60, 80, 100, 120, 140\}$ and $d_T \in \{60, 80, 100, 120, 140\}$, and the results are reported in Figure 7a and 7c. The remaining model setups reported in Section 4.4 are untouched when varying $d$ or $d_T$. We observe slightly better outcome when increasing either $d$ or $d_T$ on the industrial dataset. The patterns on Reddit and Wikipedia dataset are almost identical.

Secondly, we compare between the two methods of learning functional encoding, i.e. using flow-based model or using the non-parametric method introduced in Section 3.1. We experiment on two

---

[5]https://github.com/snap-stanford/snap/tree/master/examples/node2vec

[6]https://github.com/phanein/deepwalk

[7]https://github.com/williamleif/GraphSAGE

[8]https://github.com/PetarV-/GAT

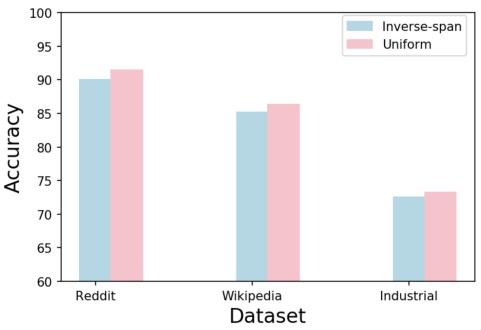

(a) Comparison between uniform and inverse timespan weighted sampling on the link prediction task

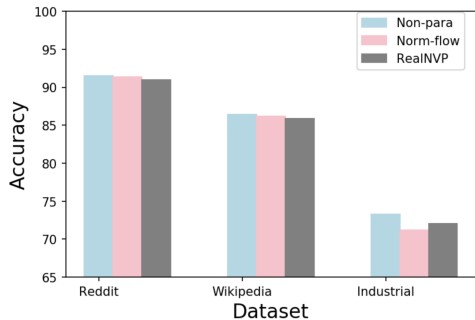

(b) Comparison between three different ways of learning the functional time encoding, on link prediction task.

Figure 6: Extra ablation study.

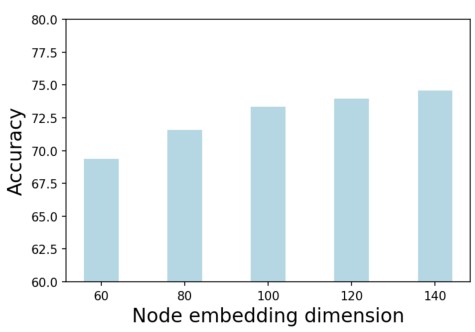

(a) Sensitivity analysis on node embeddings dimension.

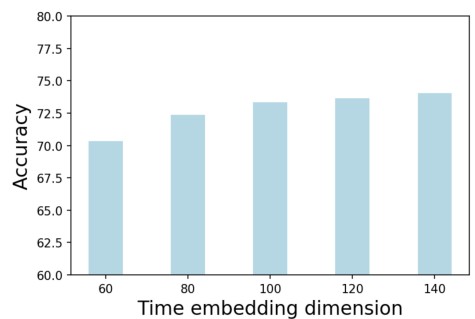

(b) Sensitivity analysis on time embeddings dimension.

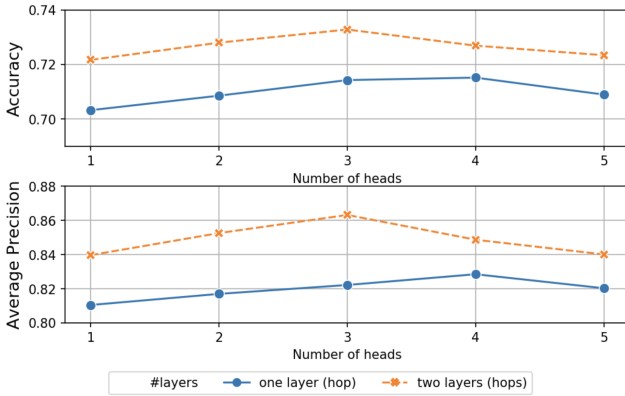

(c) Sensitivity analysis on number of attention heads and layers (hops) with $d = 100$ and $d_T = 100$.

Figure 7: Sensitivity analysis on the Industrial dataset.

flow-based state-of-the-art CDF learning method: *normalizing flow* (Rezende & Mohamed, 2015) and *RealNVP* (Dinh et al., 2016). We use the default model setups and hyper-parameters in their reference implementations[9] [10]. We provide the results in Figure 6b. As we mentioned before, using flow-based models leads to highly comparable outcomes as the non-parametric approach, but they require longer training time since they implement sampling during each training batch. However, it

---

[9]https://github.com/ex4sperans/variational-inference-with-normalizing-flows

[10]https://github.com/chrischute/real-nvp

is possible that carefully-tuned flow-based models can lead to nontrivial improvements, which we leave to the future work.

Finally, we provide sensitivity analysis on the number of attention heads and layers for *TGAT*. Recall that by stacking two layers in *TGAT* we are aggregating information from the two-hop neighbour-hood. For both *accuracy* and *AP*, using three-head attention and two-layers gives the best outcome. In general, the results are relatively stable to the number of heads, and stacking two layers leads to significant improvements compared with using only a single layer.

The ablation study for comparing between uniform neighborhood dropout and sampling with inverse timespan is given in Figure 6a. The two experiments are carried out under the same setting which we reported in Section 4.4. We see that using the inverse timespan sampling gives slightly worse performances. This is within expectation since uniform sampling has advantage in capturing the recurrent patterns, which can be important for predicting user actions. On the other hand, the results also suggest the effectiveness of the proposed time encoding for capturing such temporal patterns. Moreover, we point out that using the inverse timespan sampling slows down training, particularly for large graphs where a weighted sampling is conducted within a large number of nodes for each training batch construction. Nonetheless, inverse timespan sampling can help capturing the more recent interactions which may be more useful for certain tasks. Therefore, we suggest to choose the neighborhood dropout method according to the specific use cases.

