# OpenReview forum: "Inductive representation learning on temporal graphs"
_ICLR.cc/2020/Conference — Accept (Poster)_

### Official Review · AnonReviewer1 · 2019-10-22
**Official Blind Review #1**

**Rating:** 8

**Review:**

The major contribution of this paper is the use of random Fourier features as temporal (positional) encoding for dynamic graphs. These encodings are concatenated with standard node embeddings in transformer-like attention calculations for graph message passing. The reader finds that the proposed approach is interesting.

Experimental results are also favorable.

Concern: Whereas the use of random Fourier features (RFF) is well justified, a limitation is that it is based on a stationarity assumption. Thus, it may be less applicable to nonstationary structural changes. To cope with nonstationarity, a straightforward idea is to parameterize the temporal encoding by using neural networks rather than RFF. In the authors' approach, the RFF is in a sense parameterized, because the frequencies omega are learned. Nevertheless, the stationarity limitation persists.

Question: In the ablation study, what exactly is "the original positional encoding"? Are they learned embedding vectors? Since the authors consider continuous time rather than discrete time, how many embedding vectors are there?



**Experience Assessment:**

I have published in this field for several years.

**Review Assessment: Checking Correctness Of Derivations And Theory:**

I assessed the sensibility of the derivations and theory.

**Review Assessment: Checking Correctness Of Experiments:**

I assessed the sensibility of the experiments.

**Review Assessment: Thoroughness In Paper Reading:**

I read the paper at least twice and used my best judgement in assessing the paper.

---

> ### Author Response · Authors · 2019-11-10
> **To Official Blind Review #1**
>
> We thank the reviewer for the careful reading and valuable comments. According to the feedback, we added several additional experiments and explanations in the revised version of our paper. The added/modified contents are marked by RED fonts so that other reviewers are also aware of the changes we made.
>
> To Question “In the ablation study, what exactly is the original positional encoding? Are they learned embedding vectors?”
> The results we reported in the original submission are the learnt positional embedding vectors, where they are jointly optimized as free model parameters. In the revised paper, we also add the fixed position encoding suggested by Vaswani et al. (2017) in the ablation study (the green bar in Figure 3). We see that the fixed positional encoding is slightly outperformed by learnt positional encoding.
>
> To Question “Since the authors consider continuous time rather than discrete time, how many embedding vectors are there?”
> The time encoding approach proposed in our paper is entirely functional, which means it is a vector function of time. So the functional time encoding $\Phi(.)$ takes any time value (timespan) as input and outputs a $d_T$-dimensional vector as the representation of the time value (timespan). The functional form is presented in Eq5. The ability to handle continuous variable is a major advance of our approach compared with the vast majority of prior work on representation learning with discrete variables.
>
> Discussions on the concerns in the limitation of the stationarity assumption induced by using Fourier features.
> The reviewer raises a very good point, which touches on the implicit assumption made by our approach that we model the relative temporal information (timespan) instead of the absolute time. We agree that the absolute time can contain useful non-stationary temporal signals, such as the seasonality. Our approach does not take such perspective into consideration. The solution provided by the reviewer points out one possible direction, or we could treat the absolute time information as covariates and directly include them into the model, which we shall leave to future work.

---

> > ### Comment · AnonReviewer1 · 2019-11-11
> > **Thanks for the response**
> >
> > The authors responded satisfactorily and updated the paper with better clarity. I appreciate the job and maintain the assessment. A minor point to clarify regarding the question of the number of embedding vectors, is that I actually meant the other methods (not this paper). But since other methods do not take a functional treatment but use discrete positions, the question was meaningless in retrospect.

---

### Official Review · AnonReviewer2 · 2019-10-23
**Official Blind Review #2**

**Rating:** 6

**Review:**

Summary: This paper addresses the problem of representation learning for temporal graphs. That is, graphs where the topology can evolve over time. The contribution is a temporal graph attention (TGAT) layer aims to exploit learned temporal dynamics of graph evolution in tasks such as node classification and link prediction. This TGAT layer can work in an inductive manner unlike much prior work which is restricted to the transduction setting. Specifically, a temporal-kernel is introduced to generate time-related features, and incorporated into the self-attention mechanism. The results on some standard and new graph-structured benchmarks show improved performance vs a variety of baselines in both transduction and inductive settings.

Pros:
+ Dynamic graphs are an important but challenging data structure for many problems. Improved methods in this area are welcome.
+ Dealing with the inductive setting is an important advantage.
+ Clear performance improvements on prior state of the art is visible in both transductive+inductive settings and node+edge related tasks.

Cons+Questions:
1. Technical significance: Some theory is presented to underpin the approach, but in practice it seems to involve concatenating or adding temporal kernels element-wise to the features already used by GAT. In terms of implementation the concatenation in Eq 6 seems to be the only major change to GAT. I’m not sure if this is a major advance.
2. Insight. The presented method apparently improves on prior work by learning something about temporal evolution and exploiting it in graph-prediction tasks. But it's currently rather black-box. It would be better if some insight could be extracted about *what* this actually learns. What kind of temporal trends exist in the data that this method has learned? And how are they exploited in by the prediction tasks?
3. Writing. The English is rather flaky throughout. One particular recurring frustration is the use of the term “architect” which seems wrong. Probably “architecture” is the correct alternative.
4. Clarity of explanation. The paper is rather hard to follow and ambiguous. A few specific things that are not explained so well:
4.1. Eq 1+2 is not a sufficiently clear and self-contained recap of prior work.
4.2. Symbol d_T used at the start of Sec 3.1 seems to be used without prior definition making it hard to connect to previous Eq1+2.
4.3 The claim made about alternative approaches (Pg4) “Reparameterization is only applicable to local-scale distribution family, which is not rich enough”. Seems both too vague and unjustified.
4.4 The relationship between $t_i$ and the neighbours of the target node in Eq. 6 is not very clear.


**Experience Assessment:**

I do not know much about this area.

**Review Assessment: Checking Correctness Of Derivations And Theory:**

I did not assess the derivations or theory.

**Review Assessment: Checking Correctness Of Experiments:**

I assessed the sensibility of the experiments.

**Review Assessment: Thoroughness In Paper Reading:**

I read the paper at least twice and used my best judgement in assessing the paper.

---

> ### Author Response · Authors · 2019-11-10
> **To Official Blind Review #2**
>
> We thank the reviewer for the careful reading and valuable feedback. First of all, we apologize the typos, grammar mistakes and unclear notations. We will correct them in the final version. According to the feedback, we added several additional experiments and explanations in the revised version of our paper. The added/modified contents are marked by RED fonts so that other reviewers are also aware of the changes we made.
>
> Our response to the cons and questions are listed as below.
>
> To Q1:
> The attention mechanism employed by GAT is very different from our approach. And it is due to the different formulations that our approach works better than GAT as well as the enhanced version of GAT (GAT+T in Table 1,2,3) which operates by concatenating our time encoding to the node features. The detailed comparisons between the attention mechanism of our approach and the GAT are provided in Appendix A.2. Therefore, the major contribution of our work is the functional time encoding as well as the graph neural network architecture.
>
> To Q2:
> This is a great series of questions. Model interpretation remains to be a key challenge for deep learning models. We decide to look into the “black box” by ad-hoc model analysis on the attention weights. We refer the reviewer to the new Section 4.6 (Attention Analysis) in the revised paper for the detailed results and analysis.
>
> To Q3:
> We thank the reviewer for pointing out the improper use of “architect”. We have replaced “architect” with “architecture” in the revised version.
>
> To Q4.1:
> It is true that the introduction on self-attention in Section 2 is not self-contained since we have assumed certain background knowledge from readers. In the revised version, we provide some additional introductions in Section 2 to build more connections between prior work and our approach.
>
> To Q4.2:
> We have mentioned in the first sentence of Section 3.1 that $d_T$ is the dimension of the time encoding functional space. And since we are using time encoding to replace the positional encoding in Eq1, we have implicitly assumed that $d_T=d_{pos}$. In the revised paper, we provide additional explanations in the beginning of Section 3.1 for better clarifications.
>
> To Q4.3:
> We agree that the statement original statement on ‘reparameterization trick’ is not rigorous, and we thank the reviewer for pointing this out. In the revised paper, we replace the statement with:
> “However, the reparameterization trick is often limited to certain distributions such as the ’local-scale’ family, which may not be rich enough for our purpose. For instance, when $p(\omega)$ is multimodal it is difficult to construct the underlying distribution via direct reparameterizations.”
> Indeed, the underlying distribution of $\omega$ is unknown, so there is no way to justify if it is truly out of the range of direct reparameterization. Therefore, when selecting the appropriate distribution learning approach, we prefer models with higher complexity (larger parameter space in this case).
>
> To Q4.4:
> In Eq6, the time $t$ is the target time at which we wish to obtain the embedding, and $t_i$ is the time when the target node interacts with its neighboring node $v_i$. Therefore, $t – t_i$ is the timespan between the target time and the prior interaction time of $v_0$ (target node) and $v_i$.
>
> Finally, we once again thank the reviewer for the time and efforts in reviewing our paper. Your feedbacks are very important for us improving our work. We look forward to further comments and discussions.

---

### Official Review · AnonReviewer4 · 2019-10-28
**Official Blind Review #4**

**Rating:** 6

**Review:**

This paper proposed the temporal graph attention layer which aggregates in-hop features with self-attention and incorporates temporal information with Fourier based relative positional encoding. This idea is novel in GCN field. Experimental results demonstrate that the TGAT which adds temporal encoding outperforms the other methods. Overall this paper addressed its core ideas clearly and made proper experiments and analysis to demonstrate the superiority against existing counterparts.

There are some things need to be further answered. The baselines compared in this paper seems to be too weak. For example, how does T-GraphSage (GraphSAGE+Temporal encoding) work? How does the single-head variant of TGAT work? How does the original GAT work plus temporal encoding (as I notice TGAT uses self-attention which is similar but may not be equivalent to original GAT attention formulation, are they equivalent or not?)

**Experience Assessment:**

I have read many papers in this area.

**Review Assessment: Checking Correctness Of Derivations And Theory:**

I assessed the sensibility of the derivations and theory.

**Review Assessment: Checking Correctness Of Experiments:**

I carefully checked the experiments.

**Review Assessment: Thoroughness In Paper Reading:**

I read the paper at least twice and used my best judgement in assessing the paper.

---

> ### Author Response · Authors · 2019-11-10
> **To Official Blind Review #4**
>
> First of all, we want to thank for the reviewer for the careful reading and constructive comments. According to the feedback, we added several additional experiments and explanations in the revised version of our paper. The added/modified contents are marked by RED fonts so that other reviewers are also aware of the changes we made.
>
> The additional experiments we conducted are:
> 1.	GraphSAGE-mean + time encoding (GraphSAGE+T) by concatenating time embedding with node features for all three tasks on all datasets (Table 1,2,3 in Page 8,9);
> 2.	GAT + time encoding (GAT+T) by concatenating time embedding with node features for all three tasks on all datasets (Table 1,2,3 in Page 8,9);
> 3.	Sensitivity analysis on the number of heads and number of layers of the proposed TGAT (Figure 7c in Page 18).
>
> The relevant explanation we added according to the feedback is:
> 1.	A detailed comparison between the attention mechanism of our approach and the GAT (Appendix A.2).
>
> Our analysis on the additional experiments are provided in Section 4.3 and 4.5 in the revised paper. In general, equipping GraphSAGE and GAT with our time encoding does lead to slightly improved performances uniformly across all tasks and datasets. However, the proposed TGAT still surpass the enhanced baselines with significant margins in most cases. On one hand, the results suggest that the time encoding have potential to help extend non-temporal graph representation learning methods to temporal settings. On the other, we see that the time encoding still works the best with our network architecture which is designed for temporal graphs.
>
> The additional sensitivity analysis in Figure 7c suggests that using three attention heads with two layers gives the best performances. Using only a single head may suffer from under-fitting issues on the dataset we experimented on, since we observe increased metrics with using two and three heads. In all our experiments, we treat the number of heads as a tuning parameter, since its behavior may vary on different datasets
>
> Finally, we point out that there are significant differences between the attention mechanism employed by our approach and the GAT. The side-by-side comparisons between the two attention formulations as well as the justifications are provided in Appendix A.2.
>
> Again, we express our gratitude to the reviewer for the time and effort in reviewing our paper. Your feedbacks are very important for us improving our work. We look forward to further comments and discussions.

---

### Public Comment · ~Seyed_Mehran_Kazemi1 · 2019-09-30
**Connection to Time2Vec**

Thanks for the very interesting work!

It seems to me that the vector representation proposed for time in this work is almost identical to Time2Vec [1] (with similar motivations). The connection between Time2Vec and positional encoding has also been established in [1]. I believe the connection to Time2Vec should be highlighted in the paper. Nevertheless, combining GAT with Time2Vec for inductive representation learning on dynamic graphs is quite interesting. BTW, you may be interested in a recent survey we wrote on dynamic graphs [2].

[1] https://arxiv.org/abs/1907.05321
[2] https://arxiv.org/abs/1905.11485

---

> ### Author Response · Authors · 2019-09-30
> **Reply to connection to Time2Vec**
>
> We thank the commentary for pointing out the related work of Time2Vec. We would like to point out several fundamental differences between our proposed functional time encoding and Time2Vec.
>
> Firstly, our time encoding is motivated by the harmonic analysis and comes with solid theoretical justifications and guarantees, where Time2Vec is more heuristic-driven. For learning the functional representation of time, we refer to the classical harmonic analysis to convert the challenge of learning functional time encoding to the kernel and distributional learning problems that have been established in machine learning literature. We then prove the stochastic uniform convergence property for our proposed approach.
>
> Secondly, by couping with self-attention, we propose a whole network architecture to effectively apply the functional time encoding to learn representations on temporal graphs. By the time of our submission, there is no evidence that Time2Vec can be adapted to learn representations for temporal graphs.
>
> We want to thank the commentator for mentioning the recent survey on dynamic graphs. We will add references to several heuristic-driven time to vector approaches such as Time2Vec in our next version, and discuss on the above points upon reviewers' suggestions.

---

> > ### Public Comment · ~Seyed_Mehran_Kazemi1 · 2019-10-16
> > **Time2Vec**
> >
> > Thanks for the reply.
> >
> > The connection between Time2Vec and harmonic analysis has been discussed in Section 5.3 of [1] (Section 5.4 shows empirically that learning the frequencies from data instead of fixing them results in better performance).
> >
> > However, I agree with the authors that there currently exists no results (or methodology) on using Time2Vec for dynamic graphs and the proposed methodology is novel and interesting.

---

### Decision · Program_Chairs · 2019-12-19

**Decision:**

Accept (Poster)

**Comment:**

The major contribution of this paper is the use of random Fourier features as temporal (positional) encoding for dynamic graphs. The reviewers all find the proposed method interesting, and believes that this is a paper with reasonable contributions. One comment pointed out that the connection between Time2Vec and harmonic analysis has been discussed in the previous work, and we suggest the authors to include this discussion/comparison in the paper.